

# The concept of credible duration for the Regional Frequency Analysis including historical data: application of the FAB method to a skew storm surge database

Roberto Frau[1,2], Marc Andreewsky[1], and Pietro Bernardara[3]

[1]EDF R&D Laboratoire National d'Hydraulique et Environnement (LNHE), Chatou, 78401, France
[2]Université Paris-Est, Saint-Venant Hydraulics Laboratory (ENPC, EDF R&D, CEREMA), Chatou, 78401, France
[3]CEREA, Joint Laboratory École des Ponts ParisTech – EDF R&D, Université Paris-Est, 77455 Marne la Vallée, France

Correspondence to: Roberto Frau (*roberto.frau@edf.fr*)

**Abstract**. The design of effective coastal protections requires an adequate estimation of the annual probability of occurrence of rare events associated to a return period up to $10^3$ years. The characterisation of extreme events is carried out by statistical methods based on the extreme value theory. Usually, these methods apply to very short observations series and the uncertainties associated to high return periods are too wide to be used in a design approach. Nowadays, large and spatially distributed dataset are available and historical data approaches are providing new insight on past events. Combining local information with regional information and historical information will increase the precision and the reliability of low probability extrapolation. However, the past events data provided by single events analysis are not continuous and exhaustive as systematic records. Can we guess how exhaustive the historical records are and how many historical events have been missed by the observations? This is a very important information for the estimation of the correct empirical frequency of the extreme events. To answer these questions, the FAB method is introduced here. It is based on a new definition of the "credible duration", representing the estimate duration of the support of the data.

**Key words:** Regional Frequency Analysis; Historical data; Data archaeology; Skew storm surge; Regional credible duration; Historical duration.

## 1 Introduction

The flood risk is a noteworthy threat for anthropic and industrial settlement along the coasts. The design of adequate coastal protections requires the estimation of the probability of occurrence of extreme sea levels. In particular, a statistical evaluation of extreme storm skew surge (stochastic segment of sea level) associated to high return periods (up to 1000 years) might be necessary. Statistical methods linked to the Extreme Value Theory (Fréchet, 1928; Gnedenko, 1943; Gumbel, 1958; Coles, 2001) allow the estimation of return periods of the examined extreme variable.



In the past, the return levels' estimation has been performed for time series recorded in a single site leading to huge uncertainties. In fact, the local tide gauge's recording are usually too short (usually 30 years to 50 years long) to get reliable extreme quantiles of skew storm surge.

To have a more reliable estimation a greater amount of data is necessary. To do this a first option is to look at a regional scale. Indeed, the availability of several local datasets allowed the development of the Regional Frequency Analysis (RFA). The primary purpose of RFA is to cluster different locations in a region and to use together all the data (Cunnane, 1988; Hosking and Wallis, 1997; Van Gelder and Neykov, 1998; Bernardara et al., 2011; Weiss, 2014c). The RFA is based on the index flood principle (Dalrymple, 1960) that introduces a local index for each site preserving their peculiarities into a region. A regional distribution can be defined only after checking its statistical homogeneity (Hosking and Wallis, 1997). In particular, the probability distribution of extreme values must be the same everywhere in the region to allow the fitting of a probability distribution with a big data sample. Several applications exist for marine variables and, mostly, for skew storm surge (Bernardara et al., 2011; Bardet et al., 2011; Weiss et al., 2013).

Weiss (2014c) proposed a RFA approach to treat extreme waves and skew storm surges. According to this approach, homogeneous regions are first built relying on typical storm footprints and, their statistical homogeneity is subsequently tested. The addition of physical elements permits to get regions that are both physically and statistically homogeneous. Furthermore, this methodology and in particular the pooling method yields an equivalent duration in years that enables to determine the benefits of a regional analysis compared to a local one.

As proved in hydrology during the last 30 years, extending datasets with historical data or paleofloods is an alternative way to reduce uncertainties in the estimations of extremes (Hosking and Wallis, 1986; Stedinger and Baker, 1987; Ouarda et al., 1998; Benito et al., 2004; Payrastre et al., 2011) as well as representing better potential outliers. Recent works on local sea levels (Bulteau et al., 2014) and on local skew storm surges (Hamdi et al., 2014) pointed out that the incorporation of historical data leads to a positive benefit in an extreme frequency analysis.

The combination of these two approaches increases the amount of available data allowing getting better extreme return levels. The use of extraordinary floods events in a RFA has already been explored by (Nguyen et al., 2014) for peak discharges and, only (Hamdi et al., 2016) has explored it for oceanic and meteorological variables. In this study, a methodology to include historical skew storm surge in a Regional Frequency Analysis is proposed.

In particular, we consider as historical data are all the extreme marine events not recorded by tide gauges. Specifically, to find historical skew storm surge it has been necessary to go back in time and to look for informations during each data gap in a tide gauge series. In particular, during important storms, tide gauges may be affected by a partial or total failure in the sea levels' measurements.

Anyway, Baart et al. (2011) pointed out the hitch in recovering accurate historical information. An archaeological, or historical, skew storm surge is the result of a wide investigation of several qualitative and quantitative sources. Each historical data has its own origin and its quality has to be checked. Therefore, a critical analysis of historical data must be performed before whichever statistical approach.



Using historical data together with gauged data (i.e. systematic data) is not trivial. The extreme statistical analysis requires a knowledge of the time period of the events above an extreme threshold (Leese, 1973; Payrastre et al., 2011). For the gauged data, this time is represented by the recording period (i.e. systematic duration). Historical data are isolated data points thus the duration of the support of the observation is not defined. What happened

during the time period between two or more past extreme events remains unknown and the computation of the coverage period of historical data (i.e. historical duration) is not trivial. However, the estimation of time coverage for historical data as a key step in the estimation of extreme events (Prosdocimi, 2017).

In the past, to overcome this issues, the perception threshold concept has been introduced. The perception threshold represents the minimum value over which all the historical data are reported and documented (Payrastre et al.,

2011; Payrastre et al., 2013). The hypothesis is thus set that all the events over the perception threshold must have been recorded and the time period without records corresponds to time period without extreme events. This allow to estimate the coverage period and to indirectly assume the exhaustively of the collected historical data. The hypothesis for using this approach are quite strong. In order to relax them, in this paper, a different approach is proposed to estimate the equivalent duration of historical data. This evaluation is based on the more credible

hypothesis that the frequency of the very extreme events $\lambda$ is not changing significantly in time. In other words, the average number of extreme events should be the same (obviously this has to be carefully verified and demonstrated case by case), so we can guess whether the historical observations are enough to accept the hypothesis of exhaustively.

The main goal of this study is to develop a methodology able to introduce historical data in a Regional Frequency

Analysis of extreme gauged events.

The details of this methodology are described in section 2. An application with systematic skew storm surge given by 71 tide gauges placed in the UK's, French's and Spanish's coasts and 14 historical collected skew storm surges is shown in section 3.

## 2 The regional credible duration

### 2.1 Overview

The estimation of extreme values has been made applying statistical tools to a dataset of an extreme variable. Usually, extreme datasets are originated from series that are continuous in time and they are called as systematic. This allows to know for how long these extremes have been the biggest ones. This period is the duration of the extreme dataset. In a frequency estimation proceeding, the knowledge of the duration is a real need in order to

compute the probability of exceeding for a given return period.

Our study based on a Peaks-Over-Threshold approach is applied to databases of skew storm surges and, from now on, our extreme variables are systematic skew storm surges and historical skew storm surges. Following the hypothesis check, all concepts described below can be extended also to other extreme variables.

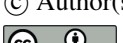


### 2.2 Basic hypothesis and credible duration for local extreme data series

The computation of the duration for systematic skew storm surges from a single site (or other systematic extreme variables) is prompt and it is equivalent to the recording period of the tide gauge from which the time series comes. The introduction of historical skew storm surge in a systematic skew storm surge dataset alters the amount of that

duration. Historical skew storm surges are isolated data points and they lead definitely to an unknown duration surplus. Therefore, they are the biggest skew storm surges occurred in an uncovered period. Fig.1 shows the different origin and nature of these two types of data. Black lines represent skew storm surge recorded by tide gauge with well-known duration. Green dotted points are the historical data and the amount of duration linked to these extreme events is unknown.

Using a Peak-Over-Threshold approach, data above a high threshold are considered as extremes. The exceedance frequency of this threshold is equivalent to the occurrence frequency of the extreme skew storm surge $\lambda$ during a defined period. This frequency $\lambda$ can be expressed as the ratio between the number of extreme skew storm surges n and the duration d of the dataset $\lambda=n/d$.

Anyway, when one or more historical skew storm surges are found and, successively, added to the local systematic

dataset, we need to know how much information we are getting including historical data in our frequency analysis. An estimation of duration based on credible hypothesis is therefore needed.

The basic hypothesis proposed here for the computation of coverage period of historical data is that the frequency of storms $\lambda$ remains unchanged in time. More precisely, $\lambda_{syst}$, namely the number of systematic skew storm surges per year over the sampling threshold, is supposed to be equal to $\lambda_{hist}$, namely the number of historical skew storm

surges per year over the same sampling threshold.

$$\lambda = \lambda_{syst} = \lambda_{hist} \qquad (1)$$

Eq.1 is in accordance with the trend lack on skew storm surges' frequency during the second part of 19[th] century, 20[th] century and the first part of 21[st] century in the regions considered in this study. The lack of a trend on skew storm surges' frequency means the storms' frequency has not a trend.

A steady frequency of storms per year is the most marked in current scientific literature. This hypothesis is supported by several references based on past observations. No significant trends are detected in terms of magnitude or frequency of storms between 1780 and 2005 in the North Atlantic and European regions since the Dalton minimum (Barring and Fortuniak, 2009). Over the past 100 years in the North Atlantic basin, no robust trends in annual numbers of tropical storms, hurricanes and major hurricanes counts have been identified by the

last IPCC report of 2013. (Hanna et al., 2008; Matulla et al., 2008; Allan et al., 2009) states that observations have no clear trends over the past century or longer with substantial decadal and longer fluctuations but with the exceptions of some regional and seasonal trends (Wang et al., 2009c; Wang et al., 2011). Moreover, MICORE (2009) in the "Review of climate change impacts on storm occurrence" affirms that results from coastal areas do not detect significant trend in frequency of storms for the existing and available datasets.

If frequency rise or fall changing occurs, the introduced concept can be adapted. For skew storm surges this hypothesis is ensured.




The assumption that the annual frequency of skew storm surges is constant in Europe during the last two centuries is widely warrantied by the several references mentioned above. A solid check must be done for others marine variables to know if they can fit this hypothesis as well.

Anyway, a new duration of observation $d_{cr}$ is defined as "credible" (i.e. based on credible hypothesis) considering a steady λ. This novelty is expressed as:

$$d_{cr} = \frac{nb_{tot}}{\lambda} = \frac{nb_{syst} + nb_{hist}}{\lambda} = \frac{nb_{syst}}{\lambda} + \frac{nb_{hist}}{\lambda} \quad (2)$$

Eq.2 shows how the number of extreme local data is divided in two sides according to the nature of data: $nb_{syst}$ is the number of extreme systematic data above the threshold u and $nb_{hist}$ is the number of extreme historical data over the same threshold u.

This splitting provides indirectly the coverage period of historical skew surges in addition to the duration of recorded data. The historical side of the duration $d_{cr}^{hist}$ can be formulated as follows:

$$d_{cr}^{hist} = \frac{nb_{hist}}{\lambda} \quad (3)$$

The coverage period on average that every historical skew surge hold is called associated duration $d_{cr,ass}$

$$d_{cr,ass} = \frac{d_{cr}^{hist}}{nb_{hist}} = \frac{1}{\lambda} \quad (4)$$

Eq.4 displays how the associated duration can be evaluated just knowing λ. Therefore, every extreme data not originated from a recorded time series has an associated duration. This is also the case for historical events occurred during a time gap in gauged skew surges series.

A coverage period for historical data can be computed thanks to the proposed hypothesis. This novelty allows to overtake the concept of perception threshold and to use it to estimate a duration for local data series.

**2.2.1 Credible duration as threshold's function**

As expressed above, $d_{cr}$ and in particular $d_{cr}^{hist}$ depend on the sampling threshold u. Eq.2 and Eq.3 show this dependence through the λ value. In order to better illustrate the concept of credible duration and its numerical change according to different thresholds, a specific focus on connection between these variables is described in this paragraph by the example shown in Figure 1.

Figure 1 shows a whole skew storm surge series in a random location. The systematic skew storm surges are represented by the continuous series of data plotted in black from about 1940 to 2017. They are provided by a tide gauge and they are observed during a fixed period equal to the recording time period, in this case, 32.88 years.

The historical skew storm surges are the green cross points obtained from different sources. As they are not obtained from continuous data series in time, their observed duration is unknown.



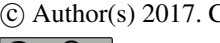

The application of the new proposed concept allows the computation of the observation's period of historical skew storm surge that represents a basic way to take into account the historical data on the estimation of extreme values. Although the period of systematic observation is known and fixed by tide gauge's records, the coverage period of historical observations is associated to the value of sampling threshold u. The hypothesis that the frequency of

storms λ has no trend leads to different values of historical duration for each λ selected value and consequently, for each likely sampling threshold. Four cases of different values of high threshold are examined.

The four cases show the behaviour of credible duration for different values of threshold. As threshold value increases, associated duration increases as well. Eq.2 expresses the credible duration as the sum of systematic duration and historical duration. Systematic duration is steady and it is independent from threshold value. For this

reason, the variation of credible duration depends only on the variation of historical duration. More the threshold is extremely high, more it is sure that no extreme skew storm surge events have been missed in the historical side of data. If the threshold is lower, then it is less certain that all extreme skew storm surge events have been observed in the historical part.

However, it is not always true that the credible duration is biggest when threshold grows (Fig.5). In fact, the

credible duration and historical duration depends not only of the threshold value but also of the number of historical events exceeding it. If the threshold is extremely high, not all historical data are taken into account in the extremes' sample. On the contrary, the associated duration of each historical data increases when threshold grows for the reasons explained in the previous paragraph (Fig.6).

Case 1 (Fig.1) uses a sampling threshold corresponding to 0.5 extremes per year (λ=0.5) and with a credible

duration of 42.88 years. A higher threshold is used in Fig.2 for the Case 2 (λ=0.23) and how you can see the credible duration grows until 54.62 years because the number of historical data over the threshold does not change. In Case 3 (Fig.3), the threshold increases (λ=0.2) but the credible duration decreases (52.88 years) because the smallest value of historical skew storm surge does not exceed the threshold. On average, the coverage period of each historical event still rises. Fig.4 (Case 4) displays the use of a very high threshold (λ=0.07) to define the

sample of extremes. In this case, only one historical event is bigger than threshold and its coverage period or, associated duration, corresponds to the historical duration. When the historical data value is higher as well the probability that the historical event covers a longer period raises.

The outlined example helps to point out the topic of this study. When the biggest extreme events are taken into account in the extreme value analysis, only few historical data are used and all the historical temporal space is

likely explored. On the contrary, the use of an extreme sample with more values leads to a lower duration caused by the non-continuous historical series.

The new expressed concept of credible duration enables to deal with historical data in a regional framework as exposed in the next paragraph.





### 2.3 Definition of the regional credible duration

Weiss (2014c) defined a methodology to fit a regional probability distribution for sea extreme variables. This approach enables to put together data from different locations and to form regions. The probability distribution of extreme skew storm surges must be the same inside a region. Thus, the extreme data of a same region are gathered

together by the use of a normalisation parameter that preserves the peculiarities of each site and the regional probability distribution is so defined. The formation of regions based on typical storm footprints (Weiss et al., 2013) is used to create physically and statistically homogeneous regions and it leads to know a master component of this regional process: the regional effective duration $D_{eff}$. This factor permits to recognise the regional profit in an extreme value analysis.

The regional effective duration $D_{eff}$ represents the number of years in which the regional sample $N_r$ is observed. $D_{eff}$ is defined as the product between the degree of regional dependence $\varphi$ $[1; N]$ and the mean of local durations

$$D_{eff} = \varphi \times \bar{d} = \frac{\lambda r}{\lambda} \bar{d} = \frac{Nr}{\lambda} \quad (5)$$

In Eq.5 $\bar{d}$ is the mean of local observations' duration of the N regional sites $\bar{d} = \sum_{i=1}^{N} d_i / N$.

The local frequency of storms $\lambda$ must be the same in all the region. This does not mean that the threshold value is the same for each site i. More details about the $D_{eff}$ are clarified in Weiss (2014b). The whole RFA approach is set for systematic data.

The need to employ historical data in the extreme analysis has been considered as highly relevant by scientific

community to get reliable quantiles and to not miss significant information. In the RFA, the use of historical records is allowed by the new notion of credible duration. A new way to process historical skew storm surges at regional scale is introduced. The addition of one or more historical skew surges in one or more sites of a region requires the knowledge of the duration of each site. Thanks to the credible duration concept applied previously to a single site, a coverage period for both types of data can be evaluated. It allows to switch from the notion of the

regional effective duration $D_{r,eff}$ to the regional *credible* duration $D_{r,cr}$:

$$D_{r,cr} = \varphi \times \bar{d} = \varphi \sum_{i=1}^{N} \frac{d_{i,cr}}{N}$$

(6)

Regional credible duration depends also on the degree of regional dependence $\varphi$ that increases when the data are

independent. Moreover, Eq.6 shows that a local duration surplus brings consequently an excess of the coverage period of regional sample. With some simplifications, the Eq.6 can thus be so reformulated:

$$D_{r,cr} = \frac{\lambda r}{\lambda} \bar{d} = \frac{Nr}{\lambda} \quad (7)$$





The lack of a trend on storms' frequency is a necessary condition to assume a steady $\lambda$. If this hypothesis is satisfied, the regional credible duration can be computed.

The regional frequency analysis of skew storm surges including historical data or, FAB method, can be performed and these two combined approaches ensure better estimations of extreme values.

### 5  3 Application of FAB method

#### 3.1 Systematic skew storm surge database and historical data

The systematic skew storm surge database is originated from time series of sea levels recorded by 71 tide gauges located along French, British and Spanish coasts of the Atlantic Ocean, the English Channel, the North Sea and the Irish Sea (Fig.7).

French data are provided by SHOM (Service Hydrographique et Océanographique de la Marine, France), English data by BODC (British Oceanographic Data Center, UK) and Spanish data by IEO (Instituto Espanol de Oceanografia, Spain). All data, except Spanish data, are updated until January 2017. Every site has different systematic durations depending on the tide gauge records. Starting from 1846, the longest time series is recorded at Brest with a duration of 156.57 years. Sea levels are hourly recorded except for British ports. BODC supplies

sea levels every 15 minutes from 1993.

Skew storm surge data are generated in each site from the difference between the maximum sea levels recorded around the time of the predicted astronomical high tide and the self-same predicted astronomical high tide. Before computing skew storm surge, sea levels must be corrected by a likely long-term alteration of mean sea levels, or eustatism. Only sea level data with significant linear trends are edited. More details on the computation of skew

storm surges are described by Simon (2007), Bernardara et al. (2011) and Weiss (2014c). In this way, the systematic database of 71 skew storm surge series is created.

Finding accurate historical data is hard and tricky and, for this reason, only 14 historical skew storm surge are used. These historical records are located in 3 of the 71 ports: 9 at La Rochelle (Gouriou, 2012; Garnier and Surville, 2010; Breilh, 2014), 4 at Dunkerque (DREAL, 2009; Parent et al., 2007; Maspataud, 2011) and 1 at

Dieppe (provided by the Service hydrographique et océanographique de la Marine SHOM, France). The historical data are introduced in corresponding skew storm surge series of the concerned sites.

This skew storm surge database of systematic and historical data is exploited in this study for the estimation of regional return periods.

#### 3.2 Formation of regions

The method to form homogeneous regions proposed by Weiss et al. (2014a) is used to get physically and statistically homogeneous regions for the database shown above. This methodology is based on a storm propagation criterion identifying the most typical storms footprints. The configuration of parameters to detect storms are set using the same criteria of Weiss (2014c). The most typical storm footprints are revealed by a Ward's





hierarchical classification (Ward, 1963) and correspond to four physical regions. The homogeneity test (Hosking and Wallis, 1997) is applied to check if all regions are also statistically homogeneous. Only one physical region is not statistically homogeneous and, after an inner partition and a further check, five physically and statistically homogeneous regions are obtained.

Moreover, a double threshold approach (Bernardara et al., 2014) is employed to separate the physical part from the statistical part of the used extreme variables and two different thresholds are used: the first to define physical storms (physical threshold) and the second one to define regional samples (statistical threshold or sampling threshold).

Fig.8 displays the five regions founded out through the application of Weiss' methodology. Although a different

input database is used, the number and the position of regions are similar to Weiss' study.

The regionalisation processing shows that only Region 1 and Region 2 hold historical data. The three sites with historical data belong actually to these two regions: La Rochelle to Region 1 and, Dieppe and Dunkerque to Region 2.

The region with more historical information is the Region 1 (nine historical skew storm surges) and the results of

FAB method proposed are focused on this region.

### 3.3 Focus on Region 1

### 3.3.1 Statistical threshold and credible duration

The sample of regional extremes is analysed with statistical tools in order to estimate regional return levels associated to very extreme return periods. A Regional Pooling Method is used to select independent maximum

data. Regional sample is formed by normalised local data. Maximum local data is divided by a local index, identified as the local sampling threshold (Weiss, 2014c). Moreover, this methodology enables to estimate the regional distribution $F_r$ considered as the distribution of the normalised maximum data above the sampling threshold.

The sampling threshold, or statistical threshold, corresponds to a storms' frequency $\lambda$ that is assumed equal in all

sites of a region. The threshold selection or, better, $\lambda$ selection is a tricky topic (Bernardara et al., 2014). There is not an univocal approach to define the exact sampling threshold. Twelve sensitivity indicators depending on all possible $\lambda$ values are proposed to assign the best $\lambda$ value for the regional analysis of extreme skew storm surge using historical data: the statistical homogeneity test (Hosking and Wallis, 1997), the stationarity test applied to storm intensities of regional sample (Hosking and Wallis, 1993), the chi-squared test for the regional distribution

(Cochran, 1952), the statistical test to detect outliers in regional sample (Hubert and Van der Veeken, 2008; Weiss, 2014c), the value and the stability of regional credible duration, the number of regional sample's data, the shape parameter's stability of the GPD distribution, the scale parameter's stability of the GPD distribution, the value of the degree of regional dependence $\varphi$, the visually look of regional return plot and the stability of local return levels. The $\lambda$ selection can not be carried out through one of these methods and only the better combination of all indicators



can enable to appoint the optimal λ. All sensitivity indicators have the same relevance in the proposed sensitivity analysis.

A skew storm surge's frequency of 0.36 is chosen for Region 1 thanks the sensitivity analysis of the statistical factors mentioned above. Taking λ=0.36 means to have on average one extreme skew storm surge over the sampling threshold every 2.78 years in each site. This threshold selection permits the computation of the credible duration for every site belonging to the Region 1. In the Region 1, all ports, except for La Rochelle, have only systematic skew storm surge and their credible duration, equal to systematic duration, is determined and independent from the sampling threshold.

On the contrary, the site of La Rochelle has nine historical skew storm surges and its credible duration depends both on systematic duration and historical duration. The systematic duration does not change and it is equal to the recording period of La Rochelle tide gauge. The historical duration differs while threshold value changes.

The credible duration at La Rochelle is 57.88 years and the contribution of historical data is almost the 43% of the total duration. Obviously, without these nine historical data, the coverage period of extreme data at La Rochelle would have been less than 25 years, equivalent to the historical duration. All the historical data are over the chosen threshold and, consequently, the associated duration of each historical data is equal to 2.77 years.

### 3.3.2 Regional credible duration

Eq.6 enables to compute the regional credible duration for Region 1. The introduction of nine historical data in the local time series and, more precisely, only eight in the regional sample after the use of Regional Pooling method, grants an earning of 11.11 years of regional duration. As Tab.2 shows, historical data increase the credible durations for sites where there are historical records and, consequently, the regional credible duration increases likewise. In Region 1, only the site of La Rochelle increases its duration. This contributes to increasing the regional credible duration.

Regional credible duration depends not only on the mean of local durations but also on the degree of regional dependence φ (Eq.6). Obviously, this degree does not change with only nine more added data. In conclusion, the rise of the regional credible duration of Region 1 is produced mostly by the increase of credible duration at La Rochelle.

Tab.2 shows the difference with the FAB method that allows to take into account historical data and a traditional method of RFA (Bernardara et al., 2011; Weiss, 2014c) that does not enable the use of historical data. Fig.9 displays how more λ decreases in value, the more the difference between regional credible duration (with the nine historical data) and the regional effective duration (without historical data) for Region 1 increases. This is due to the greater certainty that historical data above the sampling threshold are the biggest during a longer time.

### 3.3.3 Regional return levels

Regional return levels for Region 1 are obtained through the fit of a Generalized Pareto Distribution (Fig.10). Penalized maximum likelihood estimation (PMLE) (Coles and Dixon, 1999) is used to estimate the regional GPD



parameters (γ, k). As shows Fig.10, the regional distribution is unbounded due to a positive shape parameter k. The return level plot of Region 1 is produced with the 95% confidence intervals generated by a bootstrap of observed storms. A Weibull plotting position is employed for the regional empirical return period. Return levels for each site are achieved through multiplication between regional return levels and local indices, equivalent to the

local sampling threshold.

Red dotted points represent the eight historical data taken into account in Region 1. As appears, the benefit of using historical data is not only an increase of the coverage period of extreme data but also being able to consider storms that really happened in history with their intensity. The historical skew surges used in this study represent very big storms and, for this reason, it is newsworthy to take into account their large intensity. Indeed, the biggest

normalised skew storm surge considered in Region 1 is 3.01 compared with 2.13 of the regional sample generated without historical data (Tab.2).

**4 Conclusion**

The application of FAB method, based on the lack of a trend of storms' frequency λ, to a wide skew storm surge database of systematic and historical data reduces the uncertainties on the estimation of return levels linked to

extreme return periods.

The introduction of historical data on a frequency analysis of extremes is very important to increase the reliability on estimations. In the past, the unknown coverage period of historical data has usually limited the use of historical data. Historical data are not continuous in time and the real need to employ them on the extreme value analysis leads to formulate a credible hypothesis supported by several scientific studies. In this way, the credible duration

can be computed. This hypothesis is justified for skew storm surge. For other variables, a hypothesis check has to be done.

When historical data exist, the credible duration depends on sampling threshold. This new concept of duration allows to expand the credible duration's concept on regional scale. The Regional Analysis is another way to reduce uncertainties on the estimations and these two ways can be used together by the proposed FAB method.

The results describe the relevance of using historical records on a regional analysis both in terms of regional duration and of storms' intensity. Sometimes, these data can represent very large events and so we have to consider them during the estimation of extreme variables. Moreover, the existence of historical records enables to raise the coverage period of the extreme sample.

No extreme data must be missed in an extreme frequency analysis. It is very important to use all of the available

data in order to get extreme estimations increasingly more reliable and, regional analysis and historical data are two tools that can be merge with satisfactory results.

New perspectives are opened to improve extreme estimations using more historical data. Historical data are collected from different sources as ancient newspapers, archives, water marks, palaeographic studies and others. The aggregation of all of these data in an historical database is a need to exploit the maximum amount of extremes.





Furthermore, every historical data value is associated to different uncertainties and it is challenging to be able to take into account these uncertainties in the regional analysis.





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




**Table 1. Summary table of several variables involved in the computation of credible duration for 4 different cases of study.**

|  | Case 1 | Case 2 | Case 3 | Case 4 |
|---|---|---|---|---|
| Lambda λ | 0.5 | 0.23 | 0.2 | 0.07 |
| Sampling threshold (m) | 0.7 | 0.77 | 0.81 | 1.02 |
| Systematic duration (years) | 32.88 | 32.88 | 32.88 | 32.88 |
| Historical credible duration (years) | 10 | 21.74 | 20 | 14.29 |
| Number of historical data OT | 5 | 5 | 4 | 1 |
| Associated duration (years) | 2 | 4.35 | 5 | 14.29 |
| Credible duration (years) | 42.88 | 54.62 | 52.88 | 47.17 |

**Table 2. Summary table of the results obtained with the application of FAB method for Region 1.**

|  | RFA without historical data | RFA with 9 historical data (FAB, 2017) |
|---|---|---|
| Lambda λ | 0.36 | 0.36 |
| Local duration on average (years) | 41.4 | 42.36 |
| Regional credible duration (years) | 558.33 | 569.44 |
| Intensity of biggest normalized surge | 2.13 (SD) | 3.01 (HD) |

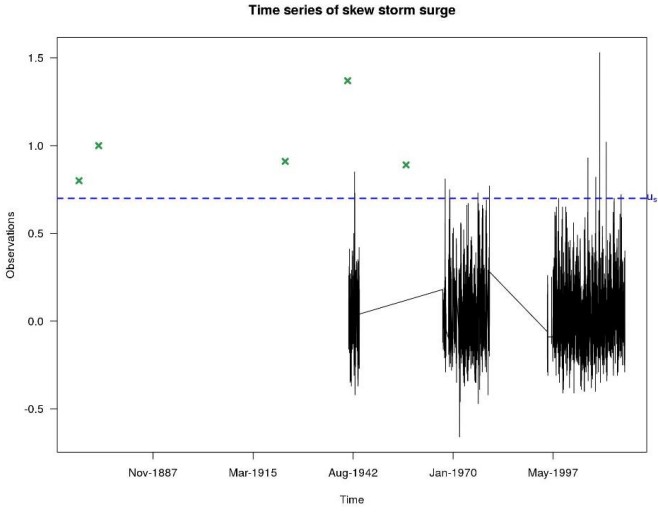

10 **Fig. 1. Fictive example of skew storm surge dataset of systematic (in black) and historical data (in green). A sampling threshold $u_s$ is used linked to a λ=0.5 (one storm over the threshold every 2 years on average)**





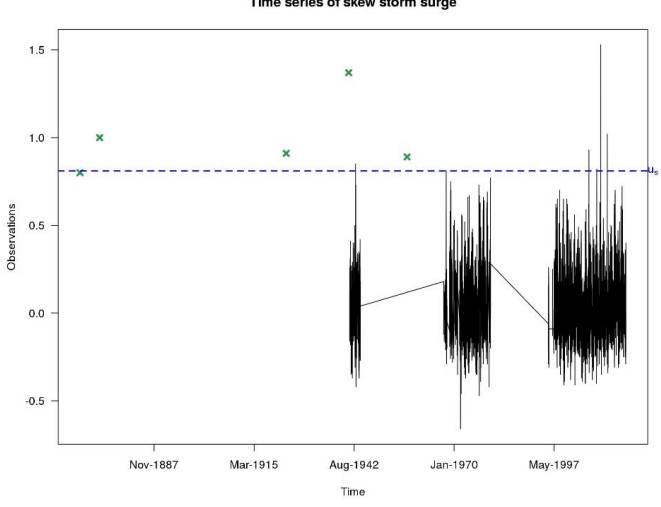

**Fig. 2.** Fictive example of skew storm surge dataset of systematic (in black) and historical data (in green). A sampling threshold $u_s$ is used linked to a $\lambda=0.23$ (one storm over the threshold every 4.35 years on average)

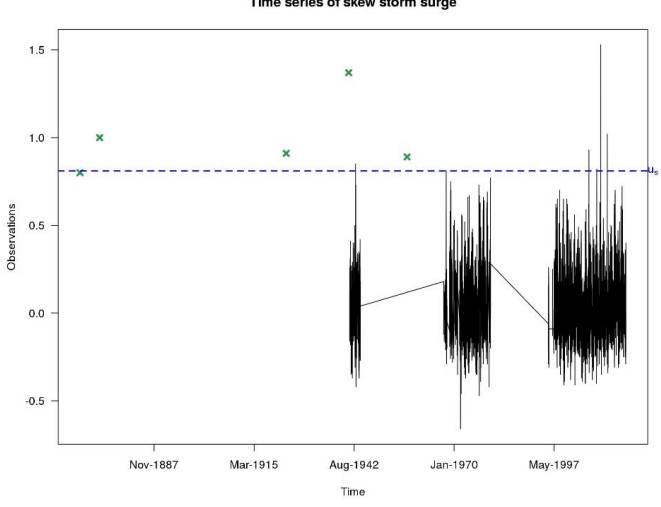

5    **Fig. 3.** Fictive example of skew storm surge dataset of systematic (in black) and historical data (in green). A sampling threshold $u_s$ is used linked to a $\lambda=0.2$ (one storm over the threshold every 5 years on average)




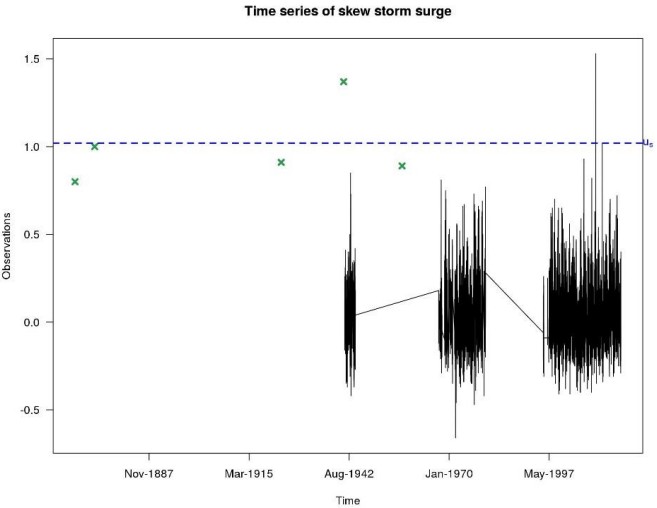

**Fig. 4. Fictive example of skew storm surge dataset of systematic (in black) and historical data (in green). A sampling threshold $u_s$ is used linked to a $\lambda=0.07$ (one storm over the threshold every 14.28 years on average)**

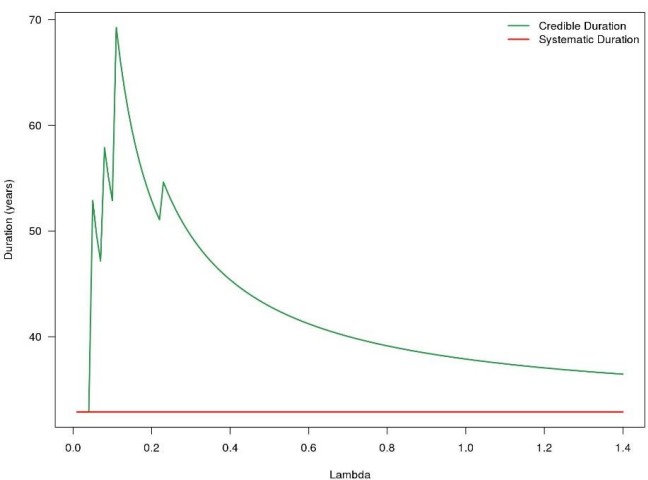

**Fig. 5. Fictive example of skew storm surge dataset of systematic and historical data - Credible duration (green line) and systematic duration (red line) for several threshold values.**




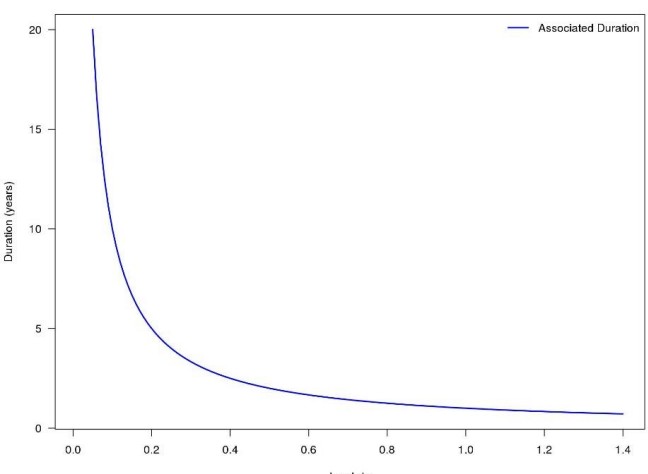

**Fig. 6.** Fictive example of skew storm surge dataset of systematic and historical data - Associated duration (blue line) for several threshold values.

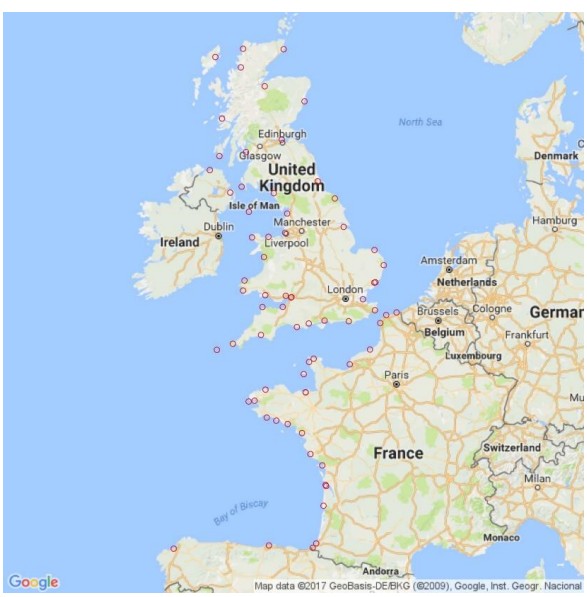

5      **Fig. 7.** Location of the 71 tide gauges used in this application.





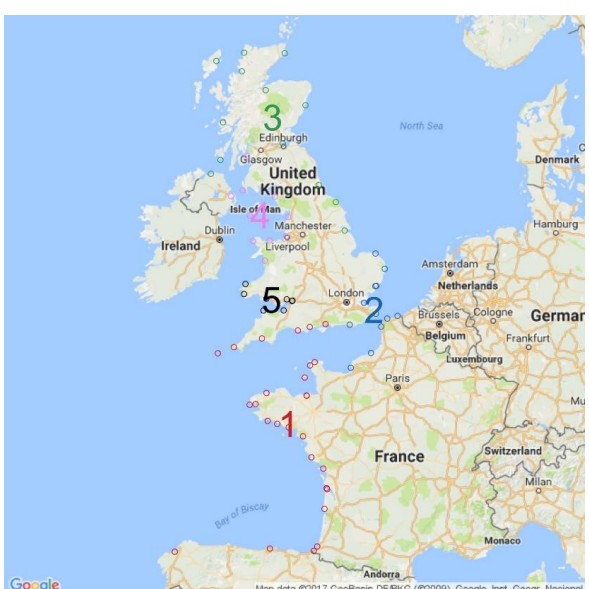

**Fig. 8. The 5 physically and statistically homogeneous regions for the 71 tide gauges.**

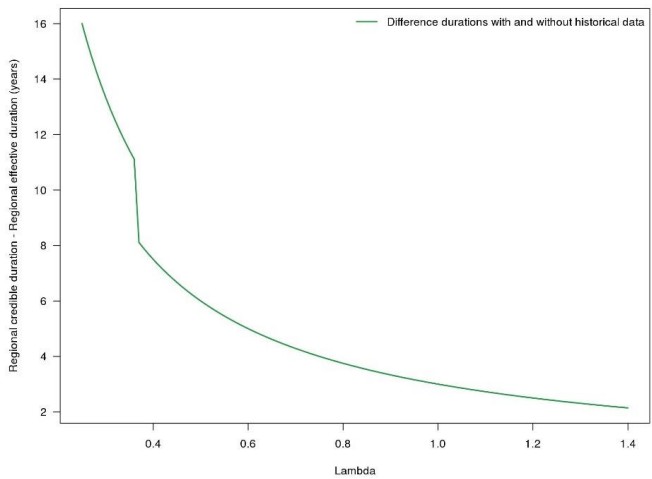

**Fig. 9. Differences between regional credible durations (taking into account the nine historical data) and regional**
5 **effective durations (taking into account only systematic data) for Region 1.**

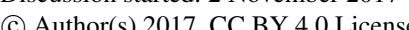



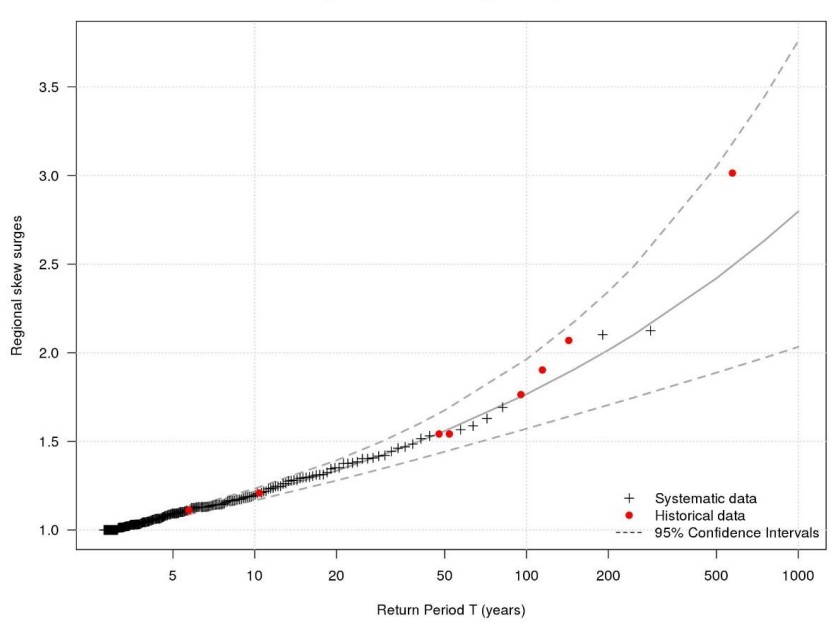

**Fig. 10. Return level plot for the regional distribution of Region 1 with the confidence intervals of 95% estimated by bootstrap. Red points represent the regional historical data and black crosses the regional systematic data.**