# Peer review of "The use of historical information on the regional frequency analysis of extreme skew surge"

_Natural Hazards and Earth System Sciences, 2017_

## Referee Comment (RC1) · Anonymous Referee #1 · 30 Nov 2017

This manuscript is clearly structured and presents an approach to reduce uncertainty associated with the estimation of return periods; an analysis often used to define events of different severity for coastal management purposes. A method to combine historical data with gauge data is presented that may be informative for other studies. The title is descriptive, but very long. If possible, could this be shortened? Below are some minor suggestion to improve the clarity of the manuscript. After revisions are made I would recommend a thorough proof read to catch remaining grammatical errors, some are listed below. The term FAB is not given in full within the manuscript what does it stand

for? Why is FAB not a keyword if this is the approach used? It must be a key method to be in the title and initiate the start of the conclusion.

In the abstract return periods up to 1000 years are mentioned, is there a reference for this level being used for rare events that can be added. The need for cost effective defence to withstand this level of event is required in the introduction. Skew storm surge can simply be referred to as skew surge throughout. However, the term needs to be defined with reference in the introduction. P1, L28, delete 'The' and start the sentence 'Flood risk is'. P2, L1, suggested updates to text: In the past, return level combinations have . . . . . .recorded at a single. . . . ... P2, L6, move 'together' from before 'all' to after 'data'. P2, L18, 'compared with a' P2, L29, 'In particular. . . ..' This sentence is unclear please reword. P2, L30-31, suggested updates to text: . . . and look for information during. . .. P2, L32, 'sea level measurements' P3, L2, replace 'whichever' with 'applying a' P3, L6, delete 'the support of' P3, L9, suggested update to text: is a key step. . .. P3 L10, this paragraph describes how data gaps in historical data suggest there were no extremes. Previously gaps in gauge data were associated to instrument failure during extremes. Ensure there is always clarity about which data is being discussed throughout the paper, historical data or gauge data. P3, L14, add 'the' estimate 'of' the coverage. P3, L14, reword: the exhaustively of the. P3, L19, 'assess' might be more appropriate than 'guess'. P3, L24, suggest updating text: gauges positioned along the UK, French and Spanish coasts. . .. P3, L29, update 'called systemic'. P3, L30, add This allows 'us' to. . .. P3, L30, Is the duration of the biggest event only considered as the duration after theat event until the next, or is if the window over that event that falls midway between this event and an earlier/later event? Please clarify. P3, 30, 'been the biggest events' P3, L33, 'exceedance'. P4, L8, suggested update to text: alters the duration of the systematic events. P4, L9, 'that occur in' P4, L11, 'tide gauge data with' P4, L25, reword 'trend lack on skew surge frequency'. It is unclear what is meant at present. P4, L26, introduce Fig 7 at this point so the reader knows the region that is being considered. P4, L27, suggested updates to text: The lack of trend in skew surge frequency also means there is a lack of trend in storm frequency. P4, L28, can

you provide a reference as an example in the literature. P4, L33, Update the brackets so the authors' names are within the sentence. P5, L5, this sentence 'If frequency . . .' is unclear. Please reword. P5, L5-10, UKCP09 should be referenced as a source of climate information for the European shelf seas. P5, L29 & P6, L9, u is not used in any equations so does not need to be defined. P6, L1, the random locations should be presented as example locations and the place names given in the figure captions. P6, L12, The four cases need to be linked to a table and the figures. P6, L15 &L16, the word 'more' needs replacing. Moreover might be more appropriate. P6, L24-32, the cases need to be introduced before L29 and Fig. 5. This information could be moved to follow the paragraph that end L5 so the four cases are introduced. P8, L11, if the RFA and FAB method are the same why is FAB used in the title when RFA is introduced/discussed in more detail. What does FAB stand for? I suggest RFA is used throughout with 1 introduction to FAB as an alternative name, giving the name in full and as the acronym. P9, L9, 'by Ward's' P9, L29, suggest updates to text: A regional sample is formed of normalised . . . are divided by. . .. P9, L30 & P10, L28, 'enables us to'. P10, L1, 'a storm's frequency' P10, L1, 'equal at all' P10, L10, 'visual look of the regional' P10, L16, 'years at each' P11, L25, The conclusion starts with the FAB method, when the RFA terminology is used more frequently in the paper. Try to consistently use 1 terminology, only indicating it could also have another name once. P11, L25, 'trend in storm frequency'. P12, L7, 'of storm frequency'. Figures 1-4, where there is no data there should be no line joining the periods of data collection. This space should be left empty to indicate no data.

---

## Referee Comment (RC2) · Anonymous Referee #2 · 7 Dec 2017

Dear editor, the manuscript in object describes a technique to use "historical data", i.e. past records of extreme events of storm surge, together with "systematic data", i.e. systematic measurements with tidal gauges, based on the concept of "credible duration", i.e. a time horizon for which extreme events are somehow recorded. The authors use this technique together with Regional Frequency Analysis, to increase the sample size for extreme value analysis. Unfortunately, though this idea may have some interesting aspects, I believe this manuscript does not meet the minimum requirements to be considered on NHESS for publication, at this stage, for different reasons. The first is that

the authors don't really discuss any reason why their technique should work better than using a perception threshold. They explain how to estimate this credible duration, but in my opinion they don't provide persuasive explanation of why this should be considered a reasonable estimation of the time horizon. They don't explain how exactly they make use of this quantity in the extreme value analysis. Last but not least, the quality of the exposition is poor: the idea of credible duration (as far as I can understand) is rather simple, but it took me quite a long of time to grasp it from explanation given in section 2. As it took me some time to undrstand that "systematic skew storm surges" means "systematic measurements of water levels". Or that "historical data" means "non-systematic records of water levels" (the term "historical data" is widely used to indicate a lot of different things, including systematic measurements and model reanalyses and hindcasts). The most positive that I can suggest is that this manuscript should be completely rewritten and resubmitted.

Follows a (sub)list of comments

The title is too long, the term "historical data" is puzzling, and I would suggest of not putting the achronym FAB here

From the abstract one does not undrestand what's really the point in this research

Abstract, line 20, .. events data provided by single event "analysis" .. do you mean .. sparse records or estimations of extreme events in absence of systematic measurements ?

Abstract, line 23. To answer these questions a new method is introduced here (hereinafter referenced as FAB, from the name of the authors)

pag 1, line 30. "stochastic segment of sea level", what do you mean? Any sea level variable can be treated as stochastic, from tides to waves.

pag 4, line 2, "duration for systematic skew storm surges". This is really unclear. Say rather "time horizon of systematic observations/measurements of ...". In general, I

would not speak about "duration", but rather "time horizon"

pag 4, line 23: "the lack of a trend on skew storm surge frequency menas the storm frequency has not a trend" this is a tautology

pag 4-5, lines 25-3: this is also a consequence of the requirements for a stationary EVA. I believe this long explanation can be simplified and shortened.

section 2: this whole explanation is not very clear. I would start explaining the concept of "credible time horizon", and then I would define it in the simplest way possible

figures 1-4: I would summarize these figures (that are almost identical) in a unique figure, trying also to explain better why they are interesting

....

---

## Author Comment (AC1) · 18 Dec 2017

Dear Referee #1,

We would like to thank for your interesting review. We are glad to see that you think that this study is a relevant approach to reduce uncertainty associated with the estimation of return periods. When we deal with coastal hazards, we cannot neglect any extreme data available. Historical data may store very important information. This methodology enables to use historical data with gauged data in order to define events of different severity for coastal management purposes.

In addition, thank you to have described our manuscript as clearly structured.

Overall, we took into account your suggestions to improve the quality and to improve even more the clarity and the form of our study.
In the following, we will give to the Referee #1 point to point detailed answer to every specific comments:

- The title is descriptive, but very long. If possible, could this be shortened?

  We agree with this helpful consideration did by Referee #1. A paper title must to provide to readers synthetically the main topic of the study. For this reason, it has been changed as follows: "The use of historical information on the regional frequency analysis of extreme skew surge". We think that this helps to understand that our work provides a suitable framework for the use of historical information; the credible duration is the innovative tool that we develop for achieving this framework. The reader will discover it reading the paper.

- The term FAB is not given in full within the manuscript what does it stand for? Why is FAB not a keyword if this is the approach used? It must be a key method to be in the title and initiate the start of the conclusion.
- P8, L11, if the RFA and FAB method are the same why is FAB used in the title when RFA is introduced/discussed in more detail. What does FAB stand for? I suggest RFA is used throughout with 1 introduction to FAB as an alternative name, giving the name in full and as the acronym.
- P11, L25, The conclusion starts with the FAB method, when the RFA terminology is used more frequently in the paper. Try to consistently use 1 terminology, only indicating it could also have another name once.

  All these helpful considerations are taken into account in the new version of the manuscript. In particular, note that the RFA (Regional Frequency Analysis) is a general term for a statistical approach using regional data. Several declinations exist in the literature, such as the Hosking and Wallis approach or the pooling approach. FAB is thus a new declination of the RFA, allowing the use of historical information.
  The term FAB is thus introduced as a keyword of this study and it is defined in the abstract (line 19). FAB is a regional method that allows to take into account not only systematic extremes but also historical extremes based on the new concept of credible duration. In order to better clarify what is FAB method, some parts are added in the new paper version as follows:

  Abstract, L20-22: "The proposed RFA method (hereinafter referenced as FAB, from the name of the authors) allows the use of historical data together with systematic data thanks to the use of the credible duration concept."

  P4, L14-15: "Thanks to this new concept of duration, in this study, the FAB methodology is developed in order to use together historical data and systematic data in a Regional Frequency Analysis of extreme events."

- In the abstract return periods up to 1000 years are mentioned, is there a reference for this level being used for rare events that can be added. The need for cost effective defence to withstand this level of event is required in the introduction.

   This is an important point that allows readers to understand why we are estimating extreme hazards for big return period. In order to provide a clear explanation, we add the reference of the French Nuclear Safety Authority to the introduction. In addition, the need of coast effective defence is outlined as follows:

   P1, L26-32: "Flood risk is a noteworthy threat for anthropic and industrial settlement along the coasts. Coast defence constructions have to withstand extreme events. Therefore, in this context, a statistical estimation of extreme marine variables (such as sea levels or skew surges, the difference between the maximum observed sea level and the maximum predicted astronomical tide level during a tidal cycle (Simon, 2007; Weiss, 2014c)) occurrence probability is necessary. These variables associated to high return periods (up to 1000 years (ASN, 2013)) are estimated by statistical methods linked to the Extreme Value Theory (Fréchet, 1928; Gnedenko, 1943; Gumbel, 1958; Coles, 2001)."

- Skew storm surge can simply be referred to as skew surge throughout. However, the term needs to be defined with reference in the introduction.

   The skew storm surge term has been replaced by skew surge throughout the article. The term has been defined with reference as follows:

   P1, L28-29 "the difference between the maximum observed sea level and the maximum predicted astronomical tide level during a tidal cycle (Simon, 2007; Weiss, 2014c)."

- P1, L28, delete 'The' and start the sentence 'Flood risk is'.

   The error has been corrected (P1, L26 of the new paper version).

- P2, L1, suggested updates to text: In the past, return level combinations have...... recorded at a single......

   The suggestion has been accepted and performed. Anyway, we are preserving the word "estimations" instead of the suggested word "combinations" that seems more suitable to this framework.

- P2, L6, move 'together' from before 'all' to after 'data'.

   The error has been corrected (P2, L3 of the new paper version).

- P2, L18, 'compared with a'.

   The sentence has been removed in the new paper version.

- P2, L29, 'In particular.....' This sentence is unclear please reword.

   The sentence has been completely reworded and better explained as follows:

"Specifically, historical data are all the skew surge events that were not recorded by tide gauge and, consequently, systematic data are all the measurements of skew surge recorded by tide gauge." (P2 L25-27 of the new paper version)

- P2, L30-31, suggested updates to text: ... and look for information during....

The suggestion has been accepted and performed (P2, L33 of the new paper version).

- P2, L32, 'sea level measurements'

The error has been corrected (P2, L34 of the new paper version).

- P3, L2, replace 'whichever' with 'applying a'

The error has been corrected (P3, L7 of the new paper version).

- P3, L6, delete 'the support of'

The error has been corrected (P3, L11-12 of the new paper version).

- P3, L9, suggested update to text: is a key step....

The suggestion has been accepted and performed (P3, L13 of the new paper version).

- P3, L10, this paragraph describes how data gaps in historical data suggest there were no extremes. Previously gaps in gauge data were associated to instrument failure during extremes. Ensure there is always clarity about which data is being discussed throughout the paper, historical data or gauge data.

This point are better discussed and clarified in the new Introduction. We defined as "historical data" are all the isolated data points derived from non-systematic measurements of skew surges. Note that this definition includes also isolated data point reconstructed during gaps of the systematic measurements. This is because statistically speaking the FAB methodology allows to treat them similar to the isolated data reconstructed from pre-gaugement period. We decided however to keep the historical data wording because this has been widely used in the literature and help focusing on the need for an effort (and the potential) in data archaeology.

P2-3, L25-2: "Specifically, historical data are all the skew surge events that were not recorded by tide gauge and, consequently, systematic data are all the measurements of skew surge recorded by tide gauge. One obvious and very promising way for collecting information not recorded by a tide gauge is to investigate and reconstruct historical events occurred before the starting date of gauge observations. This activity is also known as data archaeology and has been demonstrated very promising and useful for several study in the literature. Note however, that this definition also include isolated data point reconstructed during gaps of the systematic measurements. This is because statistically speaking they could be treated similarly to the isolated data reconstructed from pre-gaugement period. To find historical events it is thus necessary to go back in time and to look for information before the gauging period or during each data gap in a time series data. In fact, during important storms, tide gauge may be affected by a partial or total failure in the sea level measurements and so this kind of events cannot be detected. Thanks to non-systematic records, we can

hold a numerical value to these ungauged extreme events. This allows to use a bigger number of storms in the statistical assessment of the extreme skew surges.

P3, L11-14: "As historical data are isolated data points thus the duration of the observation is not defined. What happened during the time period between two or more past extreme events remains unknown. However, the coverage period estimation of historical data is a key step in the extreme value statistical models (Prosdocimi, 2017)."

- P3, L14, add 'the' estimate 'of' the coverage

The sentence has been removed in the new paper version.

- P3, L14, reword: the exhaustively of the.

The sentence has been reworded as follows:

P3, L19-21: "This allows the estimation of the coverage period and the indirect assumption that the collected historical data are exhaustive throughout the coverage period assessed."

- P3, L19, 'assess' might be more appropriate than 'guess'.

The error has been corrected (P4, L9 of the new paper version).

- P3, L24, suggest updating text: gauges positioned along the UK, French and Spanish coasts....

The suggestion has been accepted and we switch directly "placed" with "located" which seems more appropriate (P4, L16 of the new paper version).

- P3, L29, update 'called systematic'.

The sentence has been removed in the new paper version.

- P3, L30, add This allows 'us' to....

The sentence has been removed in the new paper version.

- P3, L30, Is the duration of the biggest event only considered as the duration after theat event until the next, or is if the window over that event that falls midway between this event and an earlier/later event? Please clarify.

All the concept of duration is better explained in the new paper version. This comment can be replied through this paragraph:

P5, L9-15: "When historical skew surge are available, the computation of $\lambda$ is a tricky topic. In fact, although the number of historical data $n_{hist}$ over the sampling threshold is known, we do not have any information on which happens during non-gauged period. It is thus impossible to define a duration over which the $n_{hist}$ data are observed. In other words, fixing whatever sampling threshold, we need this information for defining the frequency of occurrence of the events over the threshold (see the Eq.2). We called the missing information "historical duration" $d_{hist}$ or, in other

words, the time period for which historical skew surge above sampling threshold are estimated to be exhaustively recorded at the best of our knowledge."

- P3, 30, 'been the biggest events'

  The sentence has been removed in the new paper version.

- P3, L33, 'exceedance'.

  The sentence has been removed in the new paper version.

- P4, L8, suggested update to text: alters the duration of the systematic events

  The sentence has been removed in the new paper version.

- P4, L9, 'that occur in'

  The sentence has been removed in the new paper version.

- P4, L11, 'tide gauge data with'

  The sentence has been removed in the new paper version.

- P4, L25, reword 'trend lack on skew surge frequency'. It is unclear what is meant at present.

  The sentence has been reformulated as follows:

  P5, L23-24: "Eq.3 is in accordance with the lack of a significant trend on the extreme skew surges frequency during the second part of 19th century, 20th century and the first part of 21st century in the regions considered in this study (Fig.5)."

- P4, L26, introduce Fig 7 at this point so the reader knows the region that is being considered.

  The Fig.5 (the old Fig.7) has been introduced at the end of the sentence in brackets (P5 L12 of the new paper version).

- P4, L27, suggested updates to text: The lack of trend in skew surge frequency also means there is a lack of trend in storm frequency.

  The sentence has been removed in the new paper version.

- P4, L28, can you provide a reference as an example in the literature.

  The paragraph has been rewritten with some references as follows:

  P5, L25-27: "According to current scientific literature, based on past observations, frequency of storms has no defined trend (Hanna et al., 2008; Barring and Fortuniak, 2009; MICORE Project, 2009). Therefore, the hypothesis that storm frequency is constant has been assumed."

- P4, L33, Update the brackets so the authors' names are within the sentence.

The update has been performed (P5 L31 of the new paper version).

- P5, L5, this sentence 'If frequency...' is unclear. Please reword.

  The sentence has been reformulated as follows:

  P6, L3-6: "Therefore, the hypothesis that storm frequency is constant may be a reasonable working hypothesis for demonstrating the approach. Obviously, it is crucial to test this hypothesis before the application of the approach in the present state to any new dataset. Note that the methodology could be adapted quickly in the future for non-stationary datasets, where the introduced concept can be adjusted. For skew surges, this hypothesis is ensured".

- P5, L5-10, UKCP09 should be referenced as a source of climate information for the European shelf seas

  The UKCP09 is an excellent reference that has to be added in this paper. Thanks to Referee #1 for the suggestion.

- P5, L29 & P6, L9, u is not used in any equations so does not need to be defined.

  We define u as sampling threshold for each case shown in Section 3 and displayed in Fig.2.

- P6, L1, the random locations should be presented as example locations and the place names given in the figure captions.

  We are in according with your comment and the paragraph is rewritten as follows:

  P8, L16-21: "Fig. 1 shows the functional example of the whole skew surge series recorded at La Rochelle tide gauge (France). Systematic skew surges are represented by the continuous series of data plotted in black from about 1940 to 2017. They are observed during a fixed period equal to the recording time period, in this case, 32.88 years.
  The historical skew surges are artificial data generated by ourselves for this descriptive illustration and they are represented as green cross points. As they are not supposed to be obtained by continuous systematic records of skew surge measurements, the duration of their support is unknown."

  In addition the titles of the Figures 1,2 have been updated.

- P6, L12, The four cases need to be linked to a table and the figures.

  A link of the figures 1 and 2 of the table 1 has been added in the sentence (P8 L29-30 of the new paper version).

- P6, L15 &L16, the word 'more' needs replacing. Moreover might be more appropriate.

  The sentence has been reworded as follows:

  P9, L11-13: "In particular, when the threshold is extremely high, it is almost sure that no extreme skew surges occurred during the coverage period of historical data. If the threshold is lower, then it is less certain that all extreme skew surges have been observed in the historical part."

- P6, L24-32, the cases need to be introduced before L29 and Fig. 5. This information could be moved to follow the paragraph that end L5 so the four cases are introduced.

  The paragraph with the four cases has been moved after the paragraph that end L8. We corrected also some grammatical errors in the first sentence as follows:

  P8 L32-33: "Case 1 uses a sampling threshold $u_{C1}$ corresponding to 0.5 extremes per year ($\lambda=0.5$) and lead to a credible duration of 42.88 years (Tab.1)."

- P9, L9, 'by Ward's'

  The error has been corrected (P10, L18 of the new paper version).

- P9, L29, suggest updates to text: A regional sample is formed of normalised... are divided by....

  The suggestion has been accepted and corrected (P11, L5 of the new paper version).

- P9, L30 & P10, L28, 'enables us to'.

  The errors have been corrected (P11, L6 & P12, L2 of the new paper version).

- P10, L1, 'a storm's frequency'

  The sentence has been removed in the new paper version.

- P10, L1, 'equal at all'

  The error has been corrected (P11, L9 of the new paper version).

- P10, L10, 'visual look of the regional'

  The error has been corrected (P11, L17 of the new paper version).

- P10, L16, 'years at each'

  The error has been corrected (P11, L23 of the new paper version).

- P11, L25, 'trend in storm frequency'.

  The sentence has been removed in the new paper version.

- P12, L7, 'of storm frequency'.

  The error has been corrected as "of storm magnitude" (instead of "of storm frequency") (P13, L18 of the new paper version).

- Figures 1-4, where there is no data there should be no line joining the periods of data collection. This space should be left empty to indicate no data.

  The Figures 1-4 have been updated and the spaces you suggest have been left empty. In addition, all the figures are resumed in only one with different thresholds.

[revised manuscript text omitted]

---

## Author Comment (AC2) · 18 Dec 2017

Dear Referee #2,

We would like to thank you for the time you dedicated to our paper. We agree with a lot of your comments and we are sure that your help is adding a lot of value to our work, in particular, in the way in which the new approach is now presented.

Thank you for acknowledge that the idea to improve the RFA including historical data point and non-systematic data have some interest.

Moreover, we agree on the fact that at this stage the manuscript needs some rewritings and clarifications. The fact that the idea of credible duration is acknowledged to be rather simple, but that it took to the Referee #2 quite a long of time to grasp it from explanation given in section 2 (As for the "systematic skew storm surges" "historical data" meaning and definition) is a brilliant example of the fact that the presentation and definitions needs to be clarified. With this in mind, we largely rewrote the manuscript. We thanks the Referee #2 for pointing this out and this is what we did in the new version of the manuscript.

In the following, we will show to the Referee #2 how we took into account these general remarks and we gave point to point detailed answer to every specific comments

- The authors don't really discuss any reason why their technique should work better than using a perception threshold.

  This is a very important point, pointing directly to the first motivation of the work. We were indeed quite frustrated with the fact that the perception threshold hypothesis were constantly challenged and basically impossible to verify. In the new version of the manuscript we point it out in a new paragraph as follows:

  P3-4, L21-5: "So far, the perception threshold has allowed to use historical information in an extreme frequency analysis (Payrastre et al., 2011; Payrastre et al., 2013; Bulteau et al., 2014) but the hypothesis for using this approach seems sometimes difficult to prove.
  Indeed, the perception threshold is usually defined equal to an historical event value, based on the principles that, if any other historical event would have occurred it would have been recorded (Payrastre et al., 2011; Bulteau et al., 2014). This is a quite strong hypothesis, which is difficult to be verified and validated. Moreover, the duration associate to the perception threshold is usually defined as the duration between the oldest historical data and the start of the systematic series (Payrastre et al., 2011; Hamdi et al., 2015).
  The main point of novelty of the paper is to propose a new approach in order to deal with these questions in a more satisfactory way, the equivalent duration of historical data.
  Note that the new approach is required for the application of the RFA, at least in the POT approaches (Van Gelder and Neykov, 1998; Bernardara et al., 2011; Bardet et al., 2011; Weiss et al., 2014b). Indeed, the definition of the perception threshold equal to an historical value precludes the possibility to use RFA methodology proposed by Weiss (2014c). Specifically, in Weiss regional methodology, the number of extreme events per year $\lambda$ must be equal in each site of a region in order to estimate a regional distribution. This is impossible to achieve if the threshold is linked to the characteristic of the local observations.
  In addition, when historical information (i.e. non-systematic data) is collected during a total failure of tide gauge, this influences the frequency of the event. One could only re-estimate $\lambda$ if the credible duration is re-estimated (for details, see the section 2)."

- They explain how to estimate this credible duration, but in my opinion they don't provide persuasive explanation of why this should be considered a reasonable estimation of the time horizon.

We agree that a direct explanation on why credible duration should be considered a reasonable estimation of the time period is needed. We added in into the text. To resume we consider this new approach needed because it takes into account physical consideration (storm frequency) for the estimation of the credible duration instead of frustrating strong hypothesis on the exhaustiveness of data (such as in the perception threshold approach). For this reason, we present this in our paper as follows:

P4, L7-13: "This evaluation is based on the hypothesis that the frequency of the extreme events is not changing significantly in time. In other words, the average number of extreme events should be the same (obviously this has to be carefully verified and demonstrated case by case). In this way, we can assess the coverage period of historical data. Based on the frequency value of systematic storms occurrence, our hypothesis introduces a physical and tangible element in the estimation of the historical duration. For this reason, we state that this novelty allows to overtake the concept of perception threshold providing a credible estimation of the unknown historical duration used to estimate the extreme quantiles."

P13, L5-9: "In the past, the estimation of the period covered by the historical data collection has usually limited the use of historical data. The concept of perception threshold was employed so far to use historical data even though its hypothesis are quite strong and difficult to verify. For this reason, the new concept of credible duration has been formulated that it is based on physical considerations supported by several scientific studies. This leads to use historical isolated data points on the extreme value analysis."

- They don't explain how exactly they make use of this quantity in the extreme value analysis.

In the previous of the paper, we guessed wrongly that the use of this quantity in the extreme value analysis was implicit. However, as the Referee #2 suggests, where we make use of the credible duration quantity, needs an explicit explanation to understand the real purpose of this study. In particular, the duration of the observations are used in the EVA approach for transforming a probability of occurrence in a frequency of occurrence, introducing a time unit in the estimation.

This is exposed in the new paper version as follows:

P4, L10-30: "In the Extreme Value Analysis (EVA) of natural hazards, the estimations of extreme quantiles are performed applying statistical tools to a dataset of extreme variables. Usually, these datasets of extremes can be generated, from continuous time series data, or taking as extremes the maximum value in several predefined time blocks (Block Maxima approach) or fixing a sampling threshold (Peak-Over-Threshold approach).
Based on a POT approach, our study examines datasets of skew surges. From now on, we use skew surge as extreme variable but all concepts that we will describe can be extended also to other natural hazards.
In the POT framework, an extreme sampling threshold has to be considered in order to extract the dataset of extreme skew surges from a continuous time series data of systematic skew surges recorded by tide gauge. In this way, the extreme sample is created and the extreme quantiles computation of skew surge can be performed in

according to the statistical distribution used. The relation between return periods T and quantiles $x_T$ is expressed as:

$$P(X>x_T)=1/\lambda T \quad (1)$$

where $P(X>x_T)$ is the exceedance probability of $x_T$ or, thoroughly, the probability that in every $\lambda_T$ years an event X, strongest than $x_T$, occur. Thanks to Eq.1, the results can be illustrated through a return level plot.

The variable $\lambda$ represents the number of skew surge events per year that exceed the sampling threshold. In other words, $\lambda$ is the frequency of storms occurrence and it can be expressed as the ratio between the number of skew surges n over the sampling threshold and the recording time of systematic skew surge measurements in years d (hereinafter called systematic duration):

$$\lambda=n/d \quad (2)$$

The knowledge of n and d is necessary to compute the frequency of extreme skew surges occurrence $\lambda$ and to estimate extreme quantiles of skew surges. In other words, a duration estimation is needed to transform the probability [no units] in frequency [time$^{-1}$ dimension]."

P6-7, L31-11: "The knowledge of the regional sample duration, as exposed for a generic frequency analysis in the previously paragraph of this section, is a needed factor. In fact, the evaluation of the regional return period $T_r$ of a particular regional quantile $x_{Tr}$ is expressed as in Eq.8:

$$Tr=1/(\lambda_r \, P(X_s>x_{Tr})) \quad (8)$$

where $P(X_s>x_{Tr})$ is the probability that a storm s impacts at least one site in the region with a normalized intensity greater than $x_{Tr}$. Therefore, thanks to the regional distribution Fr, the local return period of the storm s is calculated as:

$$T=1/\lambda(1-(Fr(x_{Tr})) \quad (9)$$

In addition, Weiss (2014b) use for the computation of the empirical local return periods a Weibull plotting position. This element is correlated with the regional duration Dr:

$$T_{s,loc}=(D_r+1)/(n_r+1-rank(s)) \quad (10)$$

Then, the biggest storm $s_{max}$ of the regional sample is linked to a local return period thanks to $D_r$."

- Last but not least, the quality of the exposition is poor.

  A good quality of exposition is a key point for whatever scientific paper. An exhaustive understanding of paper concept has to be realised in the simplest way by the reader. For this reason, after thanking the Referee #2 for this useful comment, we have largely rewritten the paper and we have renovated the methodological part (Section 2).

- The title is too long, the term "historical data" is puzzling, and I would suggest of not putting the achronym FAB here

  We agree the suggestion of the Referee #2. In fact, a long and elaborate title does not tempt a reading. The title has been reduced and simplified in this way: "The use of historical information on the regional frequency analysis of extreme skew surge". The FAB acronym has been defined and replaced and the historical data has been redefined as all the isolated data points derived from non-systematic measurements of skew surges. Note that this definition includes also isolated data point reconstructed during gaps of the systematic measurements. This is because statistically speaking the FAB methodology allows to treat them similar to the isolated data reconstructed from pre-gaugement period. We decided however to keep the

historical data wording because this has been widely used in the literature and help focusing on the need for an effort (and the potential) in data archaeology.

P2-3, L25-2: "Specifically, historical data are all the skew surge events that were not recorded by tide gauge and, consequently, systematic data are all the measurements of skew surge recorded by tide gauge. One obvious and very promising way for collecting information not recorded by a tide gauge is to investigate and reconstruct historical events occurred before the starting date of gauge observations. This activity is also known as data archaeology and has been demonstrated very promising and useful for several study in the literature. Note however, that this definition also include isolated data point reconstructed during gaps of the systematic measurements. This is because statistically speaking they could be treated similarly to the isolated data reconstructed from pre-gaugement period. To find historical events it is thus necessary to go back in time and to look for information before the gauging period or during each data gap in a time series data. In fact, during important storms, tide gauge may be affected by a partial or total failure in the sea level measurements and so this kind of events cannot be detected. Thanks to non-systematic records, we can hold a numerical value to these ungauged extreme events. This allows to use a bigger number of storms in the statistical assessment of the extreme skew surges."

P3, L11-14: "As historical data are isolated data points thus the duration of the observation is not defined. What happened during the time period between two or more past extreme events remains unknown. However, the coverage period estimation of historical data is a key step in the extreme value statistical models (Prosdocimi, 2017)."

- From the abstract one does not understand what's really the point in this research

In the new version, the abstract has been almost completely rewritten in order to point out straight on the central point of this study:

"The design of effective coastal protections requires an adequate estimation of the annual occurrence probability of rare events associated to a return period up to $10^3$ years. Regional Frequency Analysis has been proved to be an applicant way to estimate extreme events using region into large and spatially distributed datasets. Nowadays, historical data are available providing new insight on past events estimations. The utilisation of historical information would increase the precision and the reliability of RFA low probability extrapolation. However, the information from past events usually looks like isolated data and they are different from continuous data from systematic measurements of tide gauges. This make the definition of the duration of our observations period. The duration of the observation period is however crucial the frequency estimation of the extreme occurrence. For this reason, we introduced here the concept of "credible duration". The proposed RFA method (hereinafter referenced as FAB, from the name of the authors) allows the use of historical data together with systematic data thanks to the use of the credible duration concept."

- Abstract, line 20, ..events data provided by single event "analysis" .. do you mean .. sparse records or estimations of extreme events in absence of systematic measurements ?

We agree that it was a very unclear sentence. Having rewritten the abstract, this sentence is not more present.

- Abstract, line 23. To answer these questions a new method is introduced here (hereinafter referenced as FAB, from the name of the authors)

This suggested part has been added in the abstract.

- pag 1, line 30. "stochastic segment of sea level", what do you mean? Any sea level variable can be treated as stochastic, from tides to waves.

The incorrect definition of skew surge has been replaced by a proper skew surge definition as follows:

P1, L26-27: "the difference between the maximum observed sea level and the maximum predicted astronomical tide level during a tidal cycle"

- pag 4, line 2, "duration for systematic skew storm surges". This is really unclear. Say rather "time horizon of systematic observations/measurements of ...". In general, I would not speak about "duration", but rather "time horizon"

This concepts has been clarified thanks to the reviewer comments. The notion of "duration for systematic skew storm surges" has been provide in the introduction part as follows:

P3, L8-11: "The extreme statistical analysis requires a knowledge of the time period for which all the events above an extreme threshold are known (Leese, 1973; Payrastre et al., 2011). For the gauged or, better, systematic data, this time is represented by the recording period (i.e. systematic duration)."

We are, however a bit more reluctant in introducing the "time horizon" wording. The point is that the terms duration is really widespread in the scientific publications, published among the other by NHESS, for the definition of the frequency of occurrence of a probabilistic variable. We worried that the introduction of a brand new wording using horizon may be confusing. Horizon is often used for the long terms climate change scenarios. It looks like more linked to the future than the pasty. Some of these studies have been referenced as follows:

  o Bulteau, T., Idier, D., Lambert, J., and Garcin, M.: How historical information can improve extreme coastal water levels probability prediction: application to the Xynthia event at La Rochelle (France), Natural Hazards and Earth System Sciences, 15, 1135-1147, doi:10.5194/nhess-15-1135-2015, 2015.
  o Hamdi, Y., Bardet, L., Duluc, C.-M., and Rebour, V.: Use of historical information in extreme surge frequency estimation: case of the marine flooding on the La Rochelle site in France, Natural Hazards and Earth System Sciences, 15, 1515-1531, doi:10.5194/nhess-15-1515-2015, 2015.
  o Sabourin, A., Renard, B.: Combining regional estimation and historical floods: A multivariate semiparametric peaks-over-threshold model with censored data, Water Resources Reseach, 51(12), doi:10.1002/2015WR017320, 2015.
  o Weiss, J., Bernardara, P., and Benoit, M.: Modeling intersite dependence for regional frequency analysis of extreme marine events, Water Resources Research, 50, 5926-5940, doi:10.1002/2014WR015391, 2014b.

- pag 4, line 23: "the lack of a trend on skew storm surge frequency means the storm frequency has not a trend" this is a tautology

During the paper rewriting, this real unclear sentence has been removed.

- pag 4-5, lines 25-3: this is also a consequence of the requirements for a stationary EVA. I believe this long explanation can be simplified and shortened.

  For a stationary EVA, the frequency and intensity stationarity is required. Intensity stationary is verified by a Student's T-test. In order to verify the frequency stationarity, we need to know the frequency of the whole sample. As Eq.2 expresses, the frequency of extremes depends on the duration value that is an unknown value (at the beginning of the study) for extreme samples with historical data. For this reason, the frequency is considered stationary as several scientific studies state. Thanks to this hypothesis, the credible duration can be estimated.

- section 2: this whole explanation is not very clear. I would start explaining the concept of "credible time horizon", and then I would define it in the simplest way possible.

  We agree with this comment. The methodological part has been re-examined, restructured and, largely rewritten in order to define the credible duration as simple as possible. Moreover, the section 2 of the new paper version begins with the explanation of the concept of credible duration, as suggested by Referee #2.

- figures 1-4: I would summerize these figures (that are almost identical) in a unique figure, trying also to explain better why they are interesting.

  The figures are summarized in a unique graph (Figure 2) where all the different thresholds are represented by different colours. The motivation of why these thresholds are interesting is expressed during all the new Section 3 and, in particular:

[revised manuscript text omitted]

---

## Author Response (AR2)

**Response to Referee #1**

Dear Referee #1,

We would like to thank you for the time you dedicated on our manuscript. We are pleased to know that you consider the revised version as greatly improved both in terms of clarity of our approach and the method used. In the revised version of the manuscript, as you kindly pointed out, the use of figures and tables within the section 3 has allowed a better explanation of our approach and a more clear definition of the credible duration.

In addition, thank you to evaluate our manuscript ready for publication following minor edits.

Your suggestions to improve the quality and the form of our study are taken into account in the last version of the manuscript and, in the following, we will give point to point detailed answer to every comments:

1) Title: 'information for regional'

   The error has been corrected.

2) Abstract: insert (RFA) after first mentioning Regional Frequency Analysis (L2).

   The acronym RFA has been inserted after first mentioning Regional Frequency Analysis.

3) P1 L31, insert (EVT) at this point and use EVT throughout manuscript (e.g., P4 L21)

   The acronym EVT has been inserted. On the other hand, the 'Extreme Value Theory' terminology is not used anymore throughout the manuscript.

4) P2 L3: RFA – the full name is required on first use followed by (RFA).

   The full name 'Regional Frequency Analysis' and its acronym have been already specified just before in **P2, L1**.

5) P2 L14: 'evaluation of the return periods of extreme events' might read better.

   The error has been corrected.

6) P2 L24: 'The proposed FAB method (a declination of RFA using credible duration) allows the estimate of reliable…'

   The suggestion has been accepted and performed.

7) P2 L34: 'During extreme storms,….and so this kind of event cannot always be detected.'

   The error has been corrected.

8) P3 L3: 'A historical event'

This grammatical error has been corrected throughout the manuscript.

9) P3 L11: 'data points, the duration'

The error has been corrected.

10) P4 L15: Use RFA.

The acronym RFA has been used.

11) P4 L21: Use EVA, the abbreviation should be introduced on P!, see comment above.

The acronym EVA has been inserted here and it is used throughout the manuscript.

12) P4 L24: add POT after Peak-Over-Threshold as POT used in the text later.

The acronym POT has been inserted.

13) P5+: Check the journal stype should variables and parameters be presented in italic in the text, e.g., d, n,…

All variables and parameters in the text are now presented in italic.

14) P7 L11: '2014b), for the generic RFA approach, which is….'

The suggestion has been accepted and performed as: '*2014b), for the classic RFA approach, which is….*'. We would like use the term "classic" rather than "generic". In fact, FAB can be considered as a more generic RFA. RFA is implemented only for systematic data and it is a more specific method. Instead, FAB method can be used also with historical data.

15) P9 sub title: 'Application of the FAB method to a real dataset'

The error has been corrected.

16) P13 L2: 'The application of the FAB method….allows more reliable ….levels to be obtained thanks to….'

The suggestion has been accepted and performed.

17) P13 L5: 'reliability of estimations'

The error has been corrected.

18) P13 L8: 'that is based on'

The error has been corrected.

The error has been corrected.

The concept of credible duration exposed and used in a local scale is expanded **to a** regional scale through the notion of regional credible duration $D_{r,cr}$. The suggestion has been performed in the new version of the manuscript.

The error has been corrected.

The suggestion has been accepted and performed.

23) P13 L24: ' estimations by using historic data that is becoming increasingly more available through different investigations of …'

The suggestion has been accepted and performed.

**Response point-by-point to Referee #2**

Dear Referee #2,

We would like to thank you again for your interesting first review that has allowed to improve a lot in our manuscript. We are glad to see that now you recommend this study for publication. The exploitation of "historical data" is a prominent way to reduce uncertainty associated with the estimation of return periods. The introduction of the new concept of "credible duration" in EVA allows satisfactorily the use of historical data in RFA.

Overall, we took into account your suggestions to still improve the quality and the clarity and the form of our manuscript.

In the following, we will give to the Referee #2 point to point detailed answer to every specific comments:

1) As I wrote in the previous review, the meaning of "historical data" is not immediately clear to readers coming from fields close to the one of the authors. For example, in many contexts the concept of "historical data" overlaps with the concept of hindcast or reanalysis. Though now an acceptable definition of historical data is given at line 25 of page 2, I would suggest the authors to clarify this also earlier, even in the abstract.

Thank you for this suggestion. Indeed, it is really important to clarify to readers the meaning of "historical data" from the beginning of this study. For this reason, we gave the definition of historical data immediately in the abstract as follows:

**Abstract, L17-18:** "*However, historical data are significant extreme events that are not recorded by tide gauge. They usually look like isolated data and they are different from continuous data from systematic measurements of tide gauges*".

2) The credible duration depends on the value of lambda that you use, and its link with the threshold. However, in section 3 it is not completely clear how you estimate lambda for your sample. Do you use only the systematic measurements for that? I would suggest to explain, indicating the formulation that you use here for lambda.

We agree that a direct explanation on how lambda value has been estimated should be provided. This direct explanation has been given in the updated version of our manuscript as follows:

**P8, L27-29:** "*The λ value is computed from the systematic measurements (Eq.2). Thanks to the hypothesis that the frequency of storms λ has no trend (Eq.3), this λ value is then used to compute the duration of the historical records. Moreover, the hypothesis of Eq.3 leads......*".

3) In the abstract: ". rfa low probability extrapolation" this sentence is unclear, please reformulate it.

This unclear sentence has been reformulated as follows:

**Abstract, L15-16:** "*The utilisation of historical information would increase the precision and the reliability of regional extreme quantiles' estimation*".

4) Line 10 of conclusions: summarize here, what this hypothesis is.

We agree with the suggestion of Referee #2 and the hypothesis has been summarize in the sentence as follows:

[revised manuscript text omitted]